# Rapid-Hardening and High-Strength Steel-Fiber-Reinforced Concrete: Effects of Curing Ages and Strain Rates on Compressive Performance

**DOI:** 10.3390/ma16144947

**Published:** 2023-07-11

**Authors:** Fan Mo, Boxiang Li, Mingyi Li, Zhuangcheng Fang, Shu Fang, Haibo Jiang

**Affiliations:** 1School of Civil and Transportation Engineering, Guangdong University of Technology, Guangzhou 510006, China; 13018556303@163.com (F.M.); lpx3199@163.com (B.L.); m15915841513@163.com (M.L.); 2Earthquake Engineering Research & Test Center, Guangzhou University, Guangzhou 510006, China; shufang@gzhu.edu.cn; 3Guangdong Key Laboratory of Earthquake Engineering & Applied Technique, Guangzhou 510006, China

**Keywords:** high-strength steel-fiber-reinforced concrete, rapid hardening, compressive performance, strain rate, curing age

## Abstract

High-strength steel-fiber-reinforced concrete (HSFRC) has become increasingly popular as a cast-in-place jointing material in precast concrete bridges and buildings due to its excellent tensile strength and crack resistance. However, working conditions such as emergency repairs and low-temperature constructions require higher demands on the workability and mechanical properties of HSFRC. To this end, a novel rapid-hardening HSFRC has been proposed, which is produced using sulphoaluminate cement (SC) instead of ordinary Portland cement. In this study, quasi-static and dynamic tests were carried out to compare the compressive behavior of conventional and rapid-hardening HSFRCs. The key test variables included SC replacement ratios, concrete curing ages, and strain rates. Test results showed: (1) Rapid-hardening HSFRC exhibited high early strengths of up to 33.14 and 44.9 MPa at the curing age of 4 h, respectively, but its compressive strength and elastic modulus were generally inferior to those of conventional HSFRC. (2) The strain rate sensitivity of rapid-hardening HSFRC was more significant compared to its conventional counterpart and increased with increasing curing ages and strain rates. This study highlights the great potential of rapid-hardening HSFRC in rapid bridge construction.

## 1. Introduction

As the service life of highway bridges increases and traffic flows become more dramatic, bridge deck pavement inevitably suffers from varying degrees of damage. A common typical disease is the breakage of expansion joints (Figure 1), which regulate the connection and displacement of the upper bridge deck structure to ensure smooth vehicle movement. It is noted that the infilled concrete is subjected to external factors such as structural shrinkage, impact fatigue, natural disasters, and vehicle overloading, which can lead to cracking, spalling of the concrete cover, and even outright failure [1,2]. High-strength steel-fiber-reinforced concrete (HSFRC) has become increasingly popular as a cast-in-place jointing material in precast assembled concrete bridges and buildings due to its excellent tensile strength and resistance to crack development [3,4]. However, the Highway Agency often requires repairs to be completed within six hours during the night so that lanes can be reopened the next morning to avoid disruption to road users [5]. However, the replacement of the existing expansion joint concrete requires a number of operations, such as cutting and chiseling of the old concrete and casting of the new concrete, which often takes a long time and can cause extensive traffic congestion. Moreover, the poor stability of the construction quality and the extremely high rework rate reduce the efficiency of the highway. The search for a rapid-hardening concrete with high early strength that can be used to repair bridge decks quickly and with minimum disruption to traffic has therefore become a high priority. For this reason, a new rapid-hardening HSFRC has been proposed, which is produced using sulphoaluminate cement (SC) instead of ordinary Portland cement [6,7]. This type of concrete is feasible in the actual construction site using rapid-paced construction and is suitable for the rapid repair of various bridge pavements. It can effectively solve the current problems of prolonged traffic closure, low traffic throughput, and economic loss due to the insufficient life of repair materials during bridge deck pavement repair.

With the increasing demand for better behavior and lower costs, mounting research has been devoted to upgrading or exploiting the structural mechanical performance by introducing innovative materials, structures, and techniques [8,9,10,11,12,13,14,15,16,17]. For conventional HSFRC, the presence of silica-fume reactive admixtures and nanoparticles is used to improve defective and porous structures within the concrete, achieving increased strength while reducing the permeability of the concrete material to chloride ions, thus protecting the internal steel fibers from corrosion [18]. In addition, fine quartz sand was used instead of river sand, and the maximum particle size was limited by not mixing coarse aggregates to improve the compactness and uniformity of the aggregate [19]. In addition, the presence of steel fibers reduces the rate of damage development in the concrete and allows for a longer yielding phase, further improving the static and dynamic strength, ductility, and toughness of HSFRC [20]. Rapid-hardening HSFRC not only retains the excellent properties of its conventional counterparts but also has four improvements [6,7]: (1) The introduction of SC with micro-expansion characteristics can completely fill the bridge deck pavement, whose early hydration can achieve rapid construction. (2) The incorporation of gypsum helps to regulate the initial setting time of concrete, which can form an encapsulation layer on the surface of particles to prevent further hydration of SC, thus enabling rapid-hardening HSFRC to achieve slow setting. (3) The addition of water-reducing agents ensures the fluidity of the fresh concrete and solves the problem that the elements cannot be formed densely due to the short setting time of SC. (4) High-temperature steam curing at 90 °C for 3 days is used to reduce material shrinkage and improve the microstructure of the material. The above improvements of rapid-hardening HSFRC ultimately result in an improvement in the three performance indicators of HSFRC, namely, rapid hardening, slow setting, and high early strength. In recent years, rapid-hardening HSFRC has been successfully adopted to repair the deteriorated pavement of a roadway in China, as shown in Figure 2. Within two hours of hardening, the lanes were reopened. After serving for three years, no visible damage was observed on the pavement (Figure 2c). This phenomenon demonstrated that the rapid-hardening HSFRC as investigated in this study exhibited favorable durability. However, the mechanical properties of rapid-hardening HSFRC still remain unclear, which hinders the application of this innovative type of concrete.

Static mechanical properties are the main performance indicators of concrete, which determine the load-bearing capacity of structures such as bridges and pavements. They are also an important basis for structural design and construction and are an essential part of the in-depth study of the dynamic mechanical properties of concrete [21,22]. Luo et al. [23] showed that the incorporation of polypropylene fibers had a significant effect on the frost resistance of HSFRC after hundreds of freeze–thaw (F–T) cycles, with a greater effect on the splitting tensile strength than on the compressive strength. Lancellotti et al. [24] used alkali-activated materials instead of Portland cement in fiber-reinforced concrete and found that the presence of fibers neither facilitated nor hindered the ground polymerization process, even though there was an increase in ionic conductivity in the samples containing fibers. As a result, a hypothesis was obtained that the samples containing fibers were less consolidated or that the dissolution of fibers contributed to the conductivity values. Vaitkevicius et al. [25] have found that a large number of microsteel fibers (up to 147 kg/m^3^) were incorporated into HSFRC to obtain excellent salt-scale resistance and favorable mechanical properties. Rady et al. [26] investigated the bond mechanism of high-strength lightweight concrete containing steel fibers with different geometries. The tested results showed that the steel fibers enhanced the internal bond strength and prevented crack extension, regardless of the geometries of the steel fibers. Furthermore, most of the existing equations for predicting tensile and bond strengths need to be modified for the case of high-strength lightweight concrete. Yu et al. [6] conducted an experimental study on HSFRC with rapid-hardening characteristics. The experimental results indicated that the 3 h strength developed fastest at 55 MPa when the gypsum substitution rate was 15%, while the later strength could be continuously increased to 81 MPa after 7 days. In addition, 2.5% (by volume) of steel fibers could increase the 3 h and 28 d compressive strength of the newly designed UHPC by 157.0% and 46.1%, respectively, compared to the reference group without fibers. Despite the above-mentioned investigations of HSFRC, few studies have focused on the static mechanical properties of rapid-hardening HSFRC.

In addition, pavements and bridges are mainly subjected to dynamic loads during services, for example, the impact friction caused by vehicles coming into direct contact with pavement, which can cause the concrete to deform and crack. Then there are the more destructive earthquakes, which can even cause bridge structures to collapse, resulting in huge losses of life and road safety. Note that the strain rates generated by impacts and earthquakes are generally higher than 100 s^−1^. When concrete is subjected to short and strong loads, its failure is considerably different from that under quasi-static loads, and the brittleness of concrete under high strain rates of impact loading is more obvious. For this reason, it is also crucial to study the dynamic mechanical properties of HSFRC. It is worth noting that in recent years there have been a large number of studies on the mechanical properties of conventional HSFRC under quasi-static loading. Murali and Vinodha [27] carried out an experimental campaign to assess the impact failure strength of steel hybrid fiber reinforced concrete (SHFRC) subjected to freezing–thawing cycles in water containing 4.0% solution of NaCl. The experimental results revealed that when the number of freezing–thawing cycles was increased, the loss in weight of SHFRC specimens was increased, and the impact failure strength of SHFRC specimens was decreased. The impact failure strength of SHFRC incorporating a higher amount of long fibers was higher compared to short fibers, which implies that long fiber played a predominant role in enhancing its impact failure strength. Li et al. [28] investigated the dynamic mechanical properties of HSFRC under the influence of freeze–thaw cycles and found that the compressive strength and energy absorption capacity of concrete gradually decreased with the increase in F–T cycles. During F–T cycles, the mechanical properties of concrete increased with the addition of steel fibers, and the optimum amount of steel fibers to enhance the resistance to F–T cycles was 1% within the evaluated range. Moein et al. [29] conducted a systematic experimental study on conventional types of HSFRC concrete cured under wet and dry conditions by drop-weight impact assessment and revealed that hooked steel fibers were more effective than crimped steel fibers in improving the impact strength, even though the length diameter was smaller. The compressive strength of concrete containing hybrid fibers (hooked + crimped) was also lower than other fibers. In addition, the moisture-cured samples had higher compressive strength (up to 12%) and tensile strength (up to 21%). Sharma et al. [30] concluded that HSFRC was effective against abrasion erosion and cavitation erosion. A systematic experimental study showed that SFRC with the addition of 1.25–1.5% steel fibers showed significant improvements in impact resistance, toughness, and energy absorption. Sun et al. [31] demonstrated that as the steel-fiber content increased, the peak stress, energy absorption, and multiple impact compressive resistance of the specimens were greatly improved. When the steel-fiber content was 6%, the dynamic impact peak strain, dynamic impact compressive strength ratio, and energy absorption capacity of the specimens were 3.09, 1.45, and 4.1 times higher than those of the reference group, respectively. Dalvand et al. [32] used zeolite material to partially replace ordinary silicate cement, which could effectively enhance the bond strength in the interfacial transition zone between fine aggregate and cement paste, thus improving the toughness and postpeak behavior. In recent years, a number of researchers have studied the mechanical properties of rapid-hardening HSFRC under quasi-static loading; however, there is limited research on its dynamic mechanical properties under impact loading [29,33]. Previous studies have shown that rapid-hardening HSFRC has good dynamic mechanical properties and its use in repairing bridge deck pavement allows a structure to dissipate more energy at the material level, reduce amplitude and stress, and improve the overall structural damping [33].

Against the above background, this paper aimed to investigate the effects of SC replacement ratios and concrete curing ages on the failure mode, ultimate condition, and stress–strain response of HSFRC under static and dynamic compression tests. It is noted that the Split Hopkinson pressure bar (SHPB) test can skillfully decouple the inertia effect in the structure and the strain rate effect in the material, which is the typical experimental technique for obtaining the dynamic compressive behavior of concrete at relatively high strain rates (10^2^~10^4^ s^−1^). Therefore, the dynamic compression tests in this study are carried out based on the SHPB experimental technique.

## 2. Experimental Program

### 2.1. Specimen Design

A number of 35 groups of (or 112) specimens were prepared and tested, including 7 groups of (or 21) cubic specimens with 100 mm sides, 7 groups (or 21) cylindrical specimens with 100 mm diameter and 200 mm height for quasi-static compression tests, and 35 groups of (or 70) flattened Brazilian discs (FBD) with 100 mm diameter and 50 mm height for dynamic compression tests. Each group of quasi-static compression specimens consisted of three nominally identical specimens and two dynamic counterparts. Two series were designed for each type of compression test. Specifically, for the quasi-static compression tests, three groups of cubic specimens and three groups of cylindrical specimens made from conventional HSFRC were tested together as control specimens to form Series I, while Series II comprised the remaining cubic and cylindrical specimens infilled with rapid-hardening HSFRC. It is noted that conventional HSFRC was introduced to better investigate the effect of SC replacement ratios on HSFRC. For the dynamic compression tests, Series I consisted of 15 groups of FBD specimens made of conventional HSFRC, while Series II consisted of 20 groups of FBD specimens made of rapid-hardening HSFRC. Each series (i.e., Series I and II in the quasi-static and dynamic compression tests) examined the influence of test parameters on the concrete curing age, using 3, 7, and 28 days for the conventional HSFRC and 4 h, 3 days, 7 days, and 28 days for the rapid-hardening HSFRC, to cover the emergency repair work opening time (4 h) and typical concrete curing ages (3, 7, and 28 days). In addition, the effect of strain rate was investigated in both Series I and II of the dynamic compression test. It should be mentioned here that the strain rate test range was determined by designing the gas pressure for the SHPB technique in the dynamic compression test. Five SHPB gas pressures were used for the dynamic compression tests, 0.5, 0.6, 0.7, 0.8, and 0.9, representing strain rates in the range of 50 to 134 s^−1^, covering a wide range of strain rates to which concrete is subjected when used in structural members. The detailed specimen arrangement for the quasi-static and dynamic compression tests is shown in Table 1. It is worth noting that instead of the quasi-static compression test where the stress and strain are directly output by the instrument and strain gauge, the average strain rate, stress, and its corresponding strain at each step (or time t) of the dynamic compression test can be obtained based on the one-dimensional stress wave propagation theory. Specifically, the average strain rate (${\dot{\varepsilon }}_{s}$), compressive strength ($\sigma_{s}$), and its corresponding strain (referred to as the ultimate strain $\varepsilon_{s}$) of all FBD specimens under dynamic compressive loading are expressed in the following equations [34,35]:(1)$${\dot{\varepsilon }}_{s}\left(t\right)=\frac{2C_{0}}{l_{s}}\varepsilon_{R}\left(t\right)$$
(2)$$\sigma_{s}\left(t\right)=\frac{E_{b}A_{b}}{A_{s}}\varepsilon_{T}\left(t\right)\;$$
(3)$$\varepsilon_{s}\left(t\right)=\frac{2C_{0}}{l_{s}}\int_{0}^{t}\varepsilon_{R}\left(t\right)dt$$
where $C_{0}$ is the speed of propagation of a one-dimensional stress wave in the SHPB; $l_{s}$ and $A_{s}$ are the thickness and cross-sectional area of FBD specimens, respectively; $\varepsilon_{R}$ is the strain of the reflected wave; and $E_{b}$ and $A_{b}$ are the elastic modulus and cross-sectional area of the SHPB, respectively.

Each specimen is given a name consisting of 3 or 4 sets of letters and/or numbers (3 sets for the quasi-static compression test and 4 sets for the dynamic compression test), plus a numeral “1, 2, or 3” to distinguish between 2 or 3 nominally identical specimens. The first set consists of the two or three byte letters “CQC, QC, or DC”, representing cubic and cylindrical specimens for quasi-static compression tests (i.e., “cubes under quasi-static compression” is shortened to “CQC” and “cylinders under quasi-static compression” is shortened to “QC”), and FBD specimens for dynamic compression tests (i.e., “FBD specimens under dynamic compression” is shortened to “DC”), respectively. The second set consists of the number “0 or 60” and the unit “%”, representing the type of infilled concrete as conventional or rapid-hardening HSFRC, respectively. The third set consists of the numbers “4, 3, 7, or 28” and the letters “h or d”, representing the curing age of the specimen as 4 h, 3 days, 7 days, or 28 days, respectively. The fourth set of specimens for dynamic compression tests only represents the gas pressure of the SHPB technique in dynamic compression tests. For example, specimen DC-60%-4h-0.5-1 represents one of two identical dynamic compressive specimens infilled with rapid-hardening HSFRC, with a curing age of 4 h and a target SHPB gas pressure of 0.5 MPa.

### 2.2. Raw Materials

The material mass proportion for both types of HSFRC was cementitious material: fine aggregate: steel fiber: water: water reducer = 42:40:6:10:2. Note that the two types of concrete differed only in the mass proportion of cementitious material, with the conventional type using ordinary Portland cement: microsilica fume: Nano-CaCO_3_ = 77:20:3, and the rapid-hardening type using sulphoaluminate concrete: ordinary Portland cement: gypsum: microsilica fume: Nano-CaCO_3_ = 45:16:16:20:3. A close-up view of the raw materials is shown in Figure 3.

### 2.3. Specimen Preparation

The same preparation procedure was used for all specimens, and the key steps are given in Figure 4. It is noted that the same type of concrete was cured in the same environment using the same methods. In particular, conventional HSFRC specimens were demolded after one day of hardening, and then placed outdoors and covered with plastic molds for one week with three dripping treatments per day. Furthermore, rapid-hardening HSFRC specimens were demolded after 2 h of hardening and subsequently steamed at 90 °C for 3 days, followed by 4 days of outdoor conditioning as with their conventional counterparts. In addition, to ensure smooth surfaces during loading to avoid stress concentrations at the specimen ends, the top and bottom surfaces of the specimens for the static compression test were covered with a high-strength plaster, while the top and bottom surfaces of the specimens for the dynamic compression test were polished with an MY259 grinder. Note that the nonparallel depth of the top and bottom surfaces of the specimens was kept below 0.02 mm [36,37].

### 2.4. Test Setup and Instrumentation

#### Quasi-Static Compression Tests

All quasi-static compression tests were carried out in accordance with the American Specification ASTM C469/C469M-14 [38]. The test setup and instrumentation are shown in Figure 5. Specifically, a displacement-controlled loading mode was used with a constant rate of 0.18 mm/min, corresponding to 10^−5^ s^−1^. All specimens were loaded in two steps: (1) Each specimen was preloaded to 10% of the expected peak load to verify loading axis alignment and proper instrument operation. (2) After preloading, all specimens were formally loaded until the load dropped to 40% of the measured peak load. In addition, the same linear displacement transducers (LVDTs) and strain gauge arrangement were used for all specimens: (1) two LVDTs installed symmetrically to cover the middle two-fifths of the column height to measure axial shortenings; (2) two strain gauges installed symmetrically in the axial direction with a gauge length of 50 mm to measure axial deformations of the midheight section in the specimen; and (3) two strain gauges installed symmetrically in the hoop direction with a gauge length of 50 mm to measure hoop deformations of the midheight section in the specimen. All test data were automatically collected every second using a data acquisition system.

All dynamic compression tests were performed in a separate SHPB apparatus in the Mechanics Laboratory of Guangdong University of Technology, China, following the Chinese Specifications GB/T 7314-2017 and GB/T 34108-2017 [39,40]. The test setup and instrumentation is shown in Figure 6. Specifically, the striker, incident, transmitted, and absorbent bars in the SHPB technique were fabricated from 60Si2Mn, which had an elastic modulus, Poisson’s ratio, mass density, and yield strength of 206 GPa, 0.3, 7740 kg/m^3^, and 1180 MPa, respectively. According to the definition of the elastic wave velocity of a bar as the square ratio of the elastic modulus to its mass density, the above-mentioned elastic wave velocity of the bar in the used SHPB technique was 5169 m/s. It was noted that wave fluctuations and dispersion would greatly affect the dynamic performance of the specimens during SHPB impact tests. Therefore, two additional measures were routinely taken to ensure uniform forces at both ends of the concrete specimen: (1) the contact surfaces of the incident bar, specimen, and transmitted bar were covered with Vaseline to reduce the frictional resistance of the contact surfaces; and (2) brass pulse shapers of 2 mm thickness and 20 mm diameter were applied between the incident bar and the specimen to retard the slope of the rising phase of the incident wave and increase the reflection time of the gravitational wave inside the specimen. In addition, four strain gauges with a gauge length of 50 mm were installed equally on the incident and transmitted bars on the SHPB technique, which were used to monitor the signals of the incident ($\varepsilon_{I}$) and reflected ($\varepsilon_{R}$) waves from the incident bar, and the transmitted ($\varepsilon_{T}$) waves from the transmitted bar, respectively.

## 3. Results and Discussion of Quasi-Static Compression Test

### 3.1. Failure Modes and Ultimate Conditions

Figure 7 shows typical failure modes of cubic and cylindrical specimens tested under quasi-static compression loading. All of the quasi-static compressive specimens failed by early crack initiation followed by rapid crack propagation and even concrete crushing. A conical final failure mode was observed in the midheight region of all specimens. It is noteworthy that as the load approached its peak, all specimens showed surface cracking of the concrete protective layer, followed by the formation of major vertical cracks and the development of multiple diagonal cracks along the vertical cracks. Furthermore, both types of HSFRC specimens showed a gradual decrease in the crack numbers as the concrete curing age increased. Furthermore, the rapid-hardening HSFRC specimens all had higher crack numbers than their conventional counterparts at the same curing ages. Note that with earlier curing ages or higher SC replacement ratios, the cubic and cylindrical specimens exhibited more significant diagonal splitting cracks. This relationship twisted with the increasing curing age or decreasing SC replacement ratio, i.e., the specimens developed a dominant failure with the vertical cracking. This finding may be related to the strength development level of the base material, with the diagonal-crack-dominated failure occurring when their diagonal shear capacity is lower than the corresponding vertical splitting capacity and, conversely, with vertical-crack-dominated failure.

Table 2 summarizes the average key test results for cubic and cylindrical specimens under quasi-static compressive loading, which include the compressive strength and its corresponding axial strain (corresponding to the ultimate axial strain), elastic modulus, and Poisson’s ratio. Specifically, the axial strain was determined by averaging the two LVDT readings installed in the midheight region of the specimen.

Figure 8 and Figure 9 show the effect of the examined parameters on the ultimate condition of quasi-static compression tests for cubic and cylindrical specimens, where all results are represented by the average of the test results for three nominally identical specimens in each test case. Note that in addition to the test results for cubic specimens shown in Figure 7, the test results for cylindrical specimens are shown in Figure 8. Figure 8 and Figure 9 show the effects of SC replacement ratios and concrete curing ages, respectively. Generally, the curing age of both types of HSFRC is proportional to the compressive strength of cubic and cylindrical specimens and inversely proportional to the cylindrical ultimate axial strain. Conversely, as the SC replacement ratio increased, the compressive strength of cylindrical HSFRC specimens at the same curing age gradually decreased while the corresponding ultimate axial strain gradually increased. It can be seen from Figure 9 that the relationship curve between the concrete curing age and compressive strength or ultimate axial strain is generally higher for specimens with conventional HSFRC than for their rapid-hardening counterparts. However, it is worth noting that the rate of strength enhancement (i.e., the slope of the curve) was significantly greater for rapid-hardening HSFRC than for conventional counterparts at the early curing age. Such a finding is understandable, as the low elastic modulus of SC makes the matrix more rigid than ordinary Portland cement, and thus exhibits a lower compressive strength at the same stress state. Nevertheless, Table 2 shows that cubic and cylindrical specimens of conventional HSFRC cured for 3 days reached 58% and 54% of those cured for 28 days, respectively, and similar comparative results were found for rapid-hardening HSFRC counterparts cured for 4 h and 28 days.

### 3.2. Stress–Strain Responses

Quasi-static compressive stress–strain curves for cylindrical specimens are shown in Figure 10 and Figure 11, each examining the effect of one test parameter. The test results show that all three nominally identical specimens have close stress–strain curves. Therefore, for ease of comparison, only one of the test results for one of the repeated specimens is given in both Figure 10 and Figure 11. Note that the positive strains in Figure 10 and Figure 11 represent the axial strains of the specimens measured by the LVDTs, while the negative strains refer to the hoop strains of the specimens measured by the strain gauges. In addition, to further illustrate the quasi-static compressive stress–strain curves for the cylindrical specimens, Figure 12 and Figure 13 each depict the effect of one test parameter on the quasi-static compressive index (including the elastic modulus and Poisson’s ratio).

Figure 10 regroups and compares the quasi-static compressive stress–strain responses of cylindrical specimens using the two different types of HSFRC. Three subplots are included in Figure 12 to illustrate the difference in quasi-static compressive performance of cylindrical specimens with two types of HSFRC for ages of 3 days, 7 days, and 28 days, respectively. It can be seen that the nonlinear growth phase of the quasi-static compressive stress–strain response was generally higher for conventional HSFRC cylindrical specimens than for their rapid-hardening HSFRC counterparts. The difference gradually increased with the increasing concrete curing age. Furthermore, Figure 12 clearly shows that the elastic modulus of HSFRC cylindrical specimens under quasi-static compression decreases with increasing SC replacement ratio; the opposite relationship was found for Poisson’s ratio and SC replacement ratio. Additionally, it can be found that rapid-hardening HSFRC cylindrical specimens had a more stable development of quasi-static compressive elastic modulus and Poisson’s ratio with the changing concrete curing age compared to their conventional counterparts.

Figure 11 illustrate the effect of concrete curing ages on the quasi-static compressive stress–strain response of cylinders with conventional and rapid-hardening HSFRC, respectively. The effect of concrete curing ages is similar to previous studies [36]: as the concrete curing age increased, the compressive strength of the cylindrical specimens under quasi-static loading increased; by contrast, the corresponding ultimate strain decreased. Furthermore, Figure 13 clearly shows that the elastic modulus and Poisson’s ratio of both types of HSFRC cylindrical specimens generally increased with increasing curing age. Interestingly, the effect of concrete curing ages on the quasi-static compressive elastic modulus and Poisson’s ratio of HSFRC cylindrical specimens was significant until the curing age was less than 7 days; however, the effect of curing age became less pronounced when the curing age was greater than 7 days. This phenomenon demonstrates that an early curing age is crucial for the development of the quasi-static compressive properties of HSFRC.

## 4. Results and Discussion of Dynamic Compression Tests

### Failure Modes and Ultimate Conditions

Figure 14 shows the typical failure modes of the FBD specimens tested under dynamic compression loading. All of the dynamic compressive specimens experienced either macroscopic cracking or crushing damage. Specifically, the concrete cracking first occurred on the sides of the FBD specimens and subsequently developed in both two-specimen planes along the height direction. Such crack propagations triggered both the spalling of the concrete protective layer and the successive appearance of crushing in the specimen core. Additionally, it was noteworthy that the specimens with both types of HSFRC exhibited more significant cracking and even crushing as the strain rate (or SHPB gas pressure) increased or the concrete curing age decreased, i.e., the integrity of the specimens after failure became increasingly degraded. Specifically, with the increasing strain rate (or SHPB gas pressure) or the decreasing concrete curing age, the specimens eventually failed in more fragments with smaller volumes. It is noted that there may be a threshold between the strain rate (or SHPB gas pressure) and specimen cracking (or crushing): when the strain rate (or SHPB gas pressure) is greater than a certain threshold, HSFRC specimens will crack or crush. Specifically, when the SHPB gas pressure is greater than 0.5 MPa (corresponding to a strain rate of 43 to 90 s^−1^), all HSFRC specimens will experience macroscopic cracking; when the SHPB gas pressure is greater than 0.8 MPa (corresponding to a strain rate of 81 to 135 s^−1^), all HSFRC specimens will experience almost complete separation of mortar, aggregate, and steel fibers. Further observations of the specimen’s failure modes were made after testing. Note that a large number of steel fibers were observed to be pulled out as a whole but not broken. This phenomenon indicates that the anchoring capacity of the steel fibers used in this study was poor, i.e., better mechanical properties of HSFRC could be achieved if steel fibers with stronger anchoring capacity were used. In addition, FBD specimens with rapid-hardening HSFRC had significantly fewer cracks than their conventional counterparts, maintaining a higher degree of postfailure integrity. This was more pronounced at the early curing age, suggesting that the introduction of SC favored the dynamic performance of the HSFRC, which may be attributed to the higher early strength.

Table 3 summarizes the key test results for all FBD specimens under dynamic compressive loading, which include the average strain rate (${\dot{\varepsilon }}_{s}$), compressive strength ($\sigma_{s}$), and ultimate strain ($\varepsilon_{s}$).

Figure 15, Figure 16 and Figure 17 show the effects of the SC replacement ratio, concrete curing age, and strain rate (or SHPB gas pressure) on the average strain rate, compressive strength, and ultimate axial strain of FBD specimens in dynamic compression tests, respectively, where all ultimate conditions are represented by the average of the test results of two nominally identical specimens in each test case. As expected, Figure 15 and Figure 17 demonstrate that for FBD specimens with different types of HSFRC, the average strain rate, dynamic compressive strength, and ultimate axial strain generally increased with increasing concrete curing age or strain rate (or SHPB gas pressure). Furthermore, Figure 15 shows that the introduction of SC enhanced the early curing age dynamic compressive performance of HSFRC specimens to some extent. Specifically, when the SHPB gas pressure was increased from 0.5 to 0.9 MPa, the mean strain rate of the specimens with conventional HSFRC increased from 43 to 132 s^−1^ and 51 to 123 s^−1^ at the concrete curing ages of 3 and 7 days, respectively. Their corresponding peak stresses increased from 35.56 to 54.79 MPa and 40.35 to 62.19 MPa, respectively, and their corresponding ultimate axial strains increased from 0.013 to 0.030 and 0.012 to 0.027, respectively. Meanwhile, the average strain rate of the specimens with rapid-hardening HSFRC increased from 70 to 122 s^−1^ and 63 to 111 s^−1^ at the concrete curing ages of 3 and 7 days, respectively. Their corresponding peak stresses increased from 39.04 to 64.53 MPa and 43.33 to 69.21 MPa, respectively, and their corresponding ultimate axial strains increased from 0.023 to 0.028 and 0.029 to 0.030, respectively. However, when the SHPB gas pressure was increased from 0.5 to 0.9 MPa at the concrete curing age of 28 days, the average strain rate of the specimens with conventional and rapid-hardening HSFRC increased from 80 to 133 s^−1^ and 80 to 126 s^−1^, respectively, and their corresponding peak stresses increased from 63.68 to 95.34 MPa and 60.13 to 99.86 MPa, respectively. In addition, their corresponding ultimate axial strains increased from 0.027 to 0.047 and 0.029 to 0.044, respectively. The above phenomenon shows that there were some differences in the dynamic compressive performance between the two types of HSFRC at the concrete curing age of 28 days, i.e., when the material properties had stabilized, but that both were generally acceptable. At the same time, rapid-hardening HSFRC has a clear advantage over its conventional counterparts in terms of dynamic compressive performance at early curing ages. These two aspects confirm the potential of rapid-hardening HSFRC for rapid bridge construction.

The dynamic compression stress–strain curve for the FBD specimen is shown in Figure 18. Note that the plot consists of 20 subplots in 5 rows and 4 pillars, where the subplots in the same row can be used to examine the effect of concrete curing ages on the dynamic compression stress–strain responses of HSFRC specimens, while their counterparts in the same pillar are used to investigate the effect of strain rate (or SHPB gas pressure). In addition, each subplot gives a comparison of specimens differing only in SC replacement ratio while other parameters remain the same, for investigating the effect of SC replacement ratio on the dynamic compression behavior of HSFRC specimens. The test results showed that both nominally identical specimens had close stress–strain curves, indicating the good repeatability of the majority of the test results. The following observations can be made from Figure 18: (1) All specimens presented comparable stress–strain responses under dynamic compression, which can be divided into four main components: a linear elastic phase, a nonlinear ascent phase, a yielding phase, and a softening phase where the specimen is completely destroyed. (2) With other parameters being the same, the longer the concrete curing age, the higher the peak point of dynamic compressive stress–strain responses of FBD specimens, but at the same time leading to a shorter yielding stage (i.e., with a constant stress but large change in strain). (3) When other parameters remained consistent, the dynamic compressive stress–strain response of specimens with rapid-hardening HSFRC was generally higher than that of their conventional counterparts when the concrete curing age was less than 7 days, but the opposite relationship existed when the curing age was greater than 7 days. (4) The dynamic compressive stress–strain response of specimens with rapid-hardening HSFRC exhibited an overall shorter yielding phase compared to their conventional counterparts. (5) The higher the strain rate (or SHPB gas pressure), the higher the dynamic compression stress–strain response of the FBD specimen when all other parameters are kept consistent.

## 5. Conclusions

This paper presents an experimental study of the quasi-static and dynamic compressive behavior of conventional and rapid-hardening HSFRCs. The key parameters examined in this study include concrete curing ages (from 4 h to 28 days) and strain rates (from 43 to 133 s^−1^). Based on the experimental results and discussion presented in this paper, the following main highlights, main findings, limitations, and areas of future research can be drawn.


**
*Main highlights and findings include the following:*
**
(1)Rapid-hardening HSFRC exhibited high early strength characteristics: the quasi-static compressive strengths of the cubes and cylinders at 4 h of curing age reached 45.16 and 33.14 MPa, respectively, which were 55 and 48% of their corresponding 28-day age strengths; in addition, the FBD specimen at 4 days of curing age had a dynamic compressive strength of up to 43.97 MPa, roughly 0.4 times their corresponding 28-day age strength.(2)The introduction of SC resulted in an increase in the strain rate sensitivity of HSFRC, where the strain rate sensitivity of rapid-hardening HSFRC increased with the increasing curing age and strain rate.(3)The compressive strength, elastic modulus, and Poisson’s ratio of rapid-hardening HSFRC were inferior to those of their conventional counterparts in both quasi-static and dynamic compressive loading, but rapid-hardening HSFRC’s postpeak deformation capacity was significantly better than that of the conventional counterparts.(4)The elastic modulus and Poisson’s ratio of HSFRC with different cement types under quasi-static compression increased with increasing curing age; the negative relationship between the elastic modulus and curing ages or strain rates was found under dynamic counterparts.


***Limitations and areas of future research include the following:*** Overall, rapid-hardening HSFRC has great potential for rapid bridge construction. In addition, the objective of this research was to verify the application feasibility of rapid-hardening HSFRC in rapid bridge construction through experimental investigations. To this end, this research only investigated the short-term compressive performance of a specific rapid-hardening HSFRC under normal conditions, lacking consideration of the effects of different material compositions, fiber types, and environmental conditions. Furthermore, to fully assess the properties of this innovative type of HSFRC, additional investigations should also be supplemented to determine the skin, the durability under the action of water and frost, the alkaline reactivity of HSFRC concrete with new cement, and so on. Moreover, the quasi-static and dynamic compressive constitutive models of this new type of concrete were unavailable due to the lack of test data on their compressive performance. Therefore, further research in this area is needed to investigate more systematically the compressive performance and mechanical modelling for the engineering application of such novel concrete.

## Figures and Tables

**Figure 1 materials-16-04947-f001:**
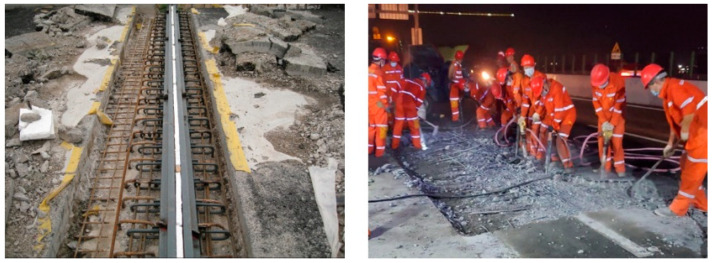
Expansion joints and corresponding repair work in bridge engineering.

**Figure 2 materials-16-04947-f002:**
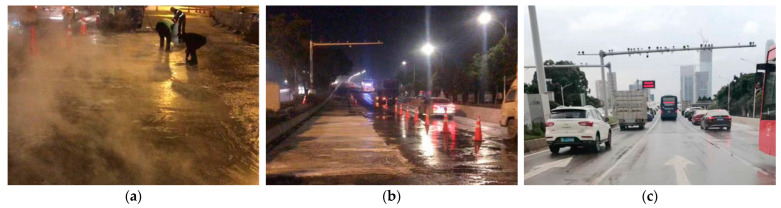
Applications of rapid-hardening HSFRC in practices. (**a**) Casting of concrete. (**b**) Hardening of concrete. (**c**) Operation situation for 3 years.

**Figure 3 materials-16-04947-f003:**
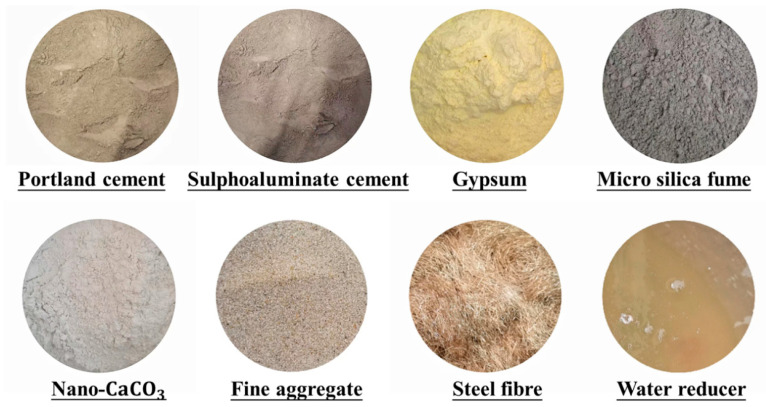
Close-ups of raw materials.

**Figure 4 materials-16-04947-f004:**
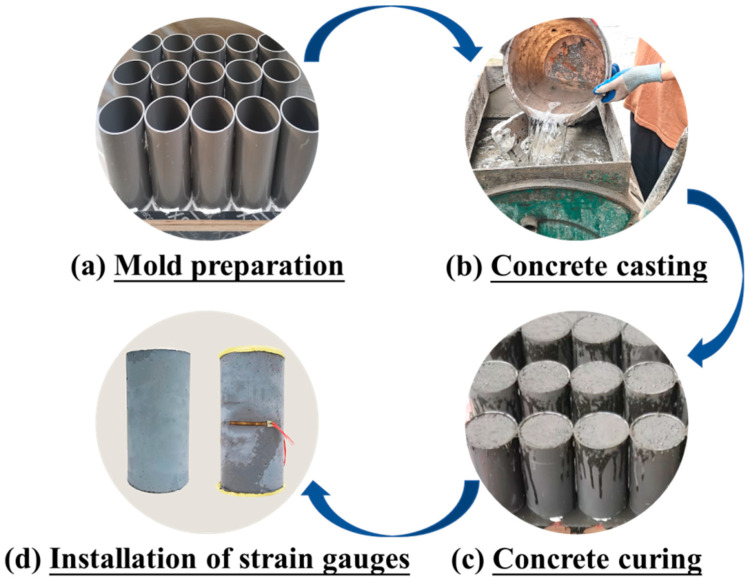
Schematic diagram of specimen preparation.

**Figure 5 materials-16-04947-f005:**
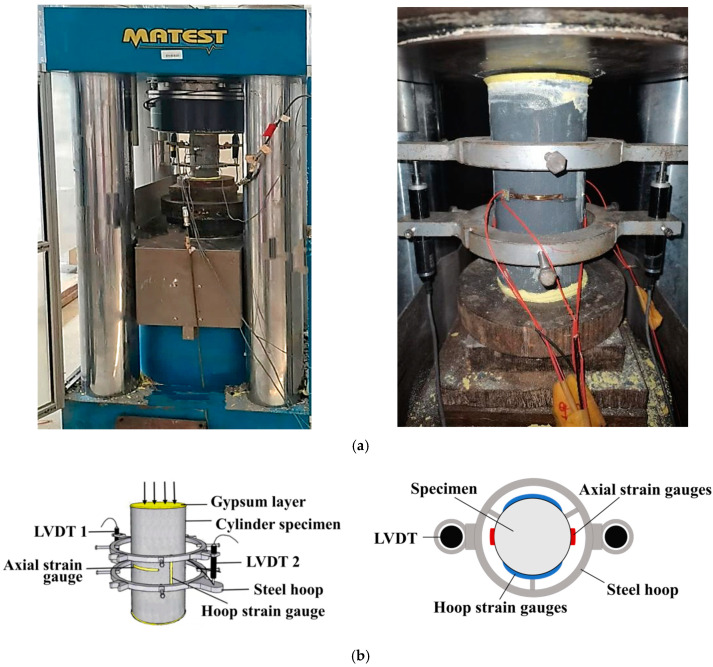
Setup and instruments of quasi-static compression tests. (**a**) Test setup. (**b**) Layout of strain gauges and LVDTs.

**Figure 6 materials-16-04947-f006:**
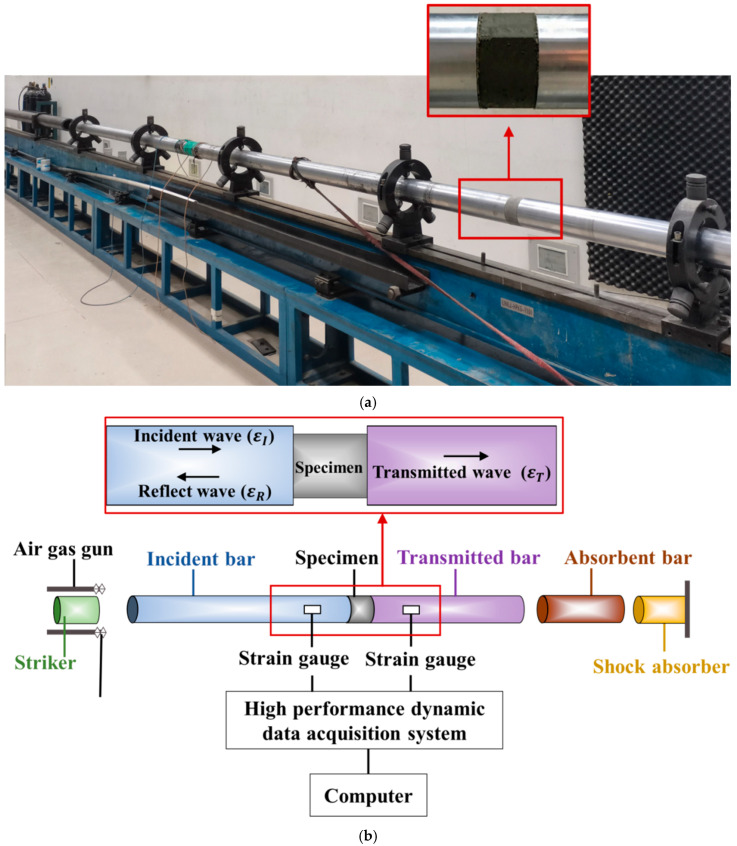
Setup and instruments of dynamic compression tests. (**a**) Test setup. (**b**) Layout of strain gauges.

**Figure 7 materials-16-04947-f007:**
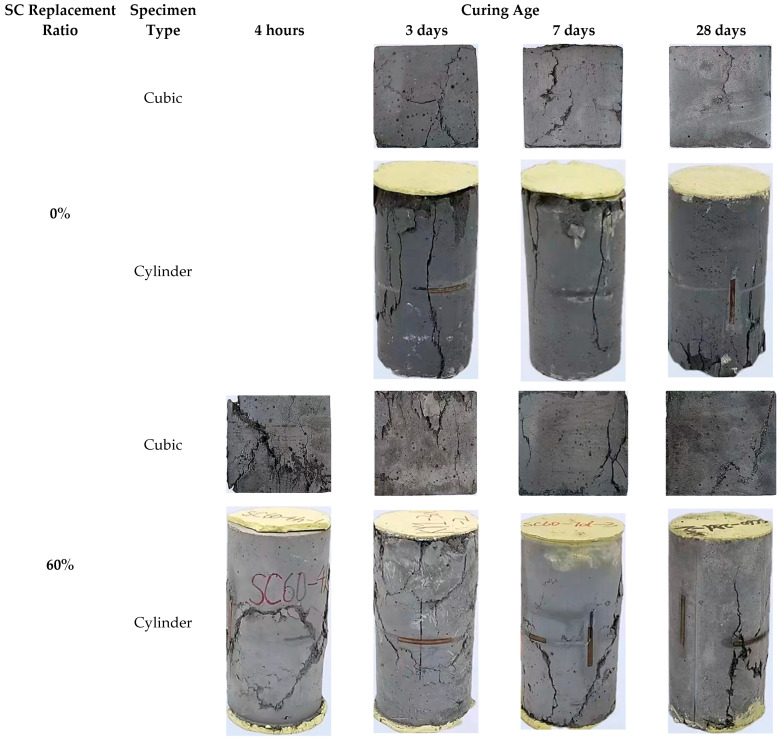
Typical failure modes of HSFRC specimens under quasi-static compression tests.

**Figure 8 materials-16-04947-f008:**
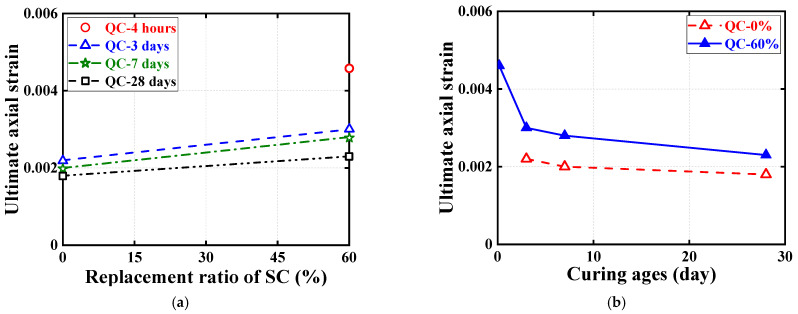
Effect of SC replacement ratios on ultimate conditions of quasi-static compression tests. (**a**) Ultimate axial stress. (**b**) Ultimate axial strain.

**Figure 9 materials-16-04947-f009:**
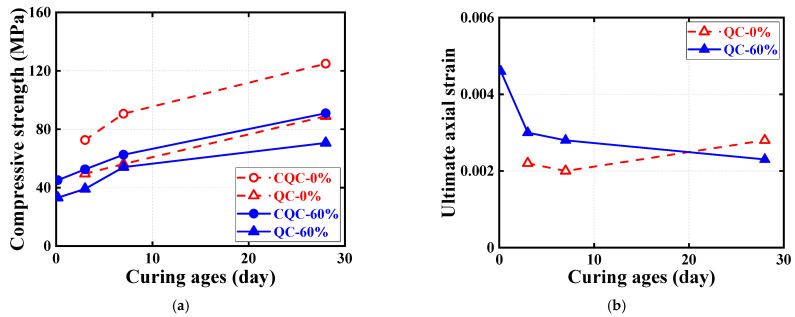
Effect of concrete curing ages on ultimate conditions of quasi-static compression tests. (**a**) Ultimate axial stress. (**b**) Ultimate axial strain.

**Figure 10 materials-16-04947-f010:**
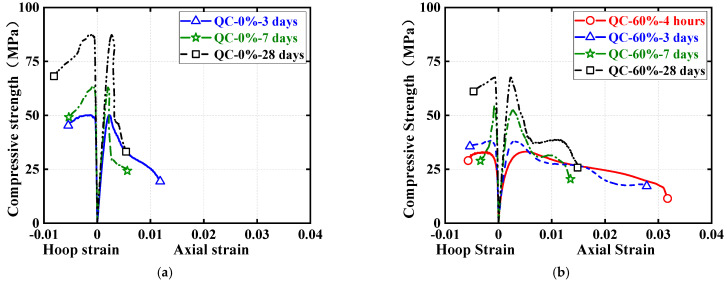
Effect of SC replacement ratios on stress–strain curves of quasi-static compression tests. (**a**) Conventional HSFRC. (**b**) Rapid-hardening HSFRC.

**Figure 11 materials-16-04947-f011:**
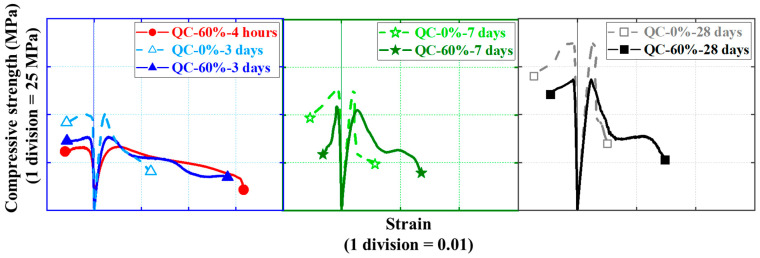
Effect of concrete curing ages on stress–strain curves of quasi-static compression tests.

**Figure 12 materials-16-04947-f012:**
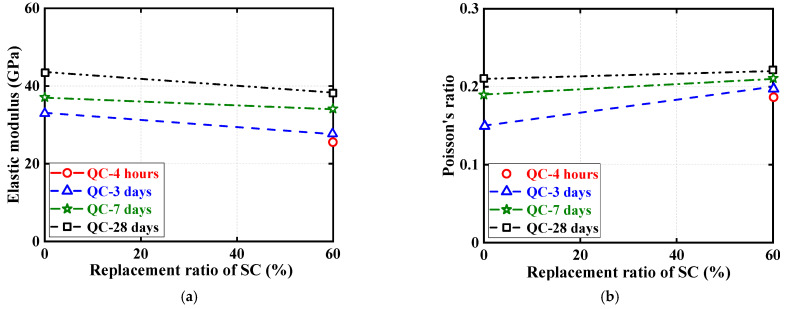
Effect of SC replacement ratios on ultimate conditions of quasi-static compression tests. (**a**) Elastic modulus. (**b**) Poisson’s ratio.

**Figure 13 materials-16-04947-f013:**
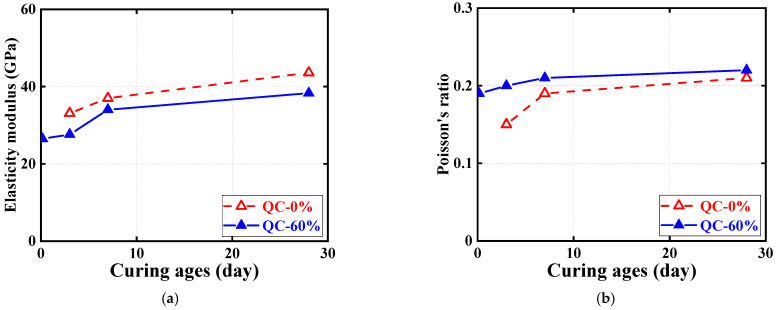
Effect of concrete curing ages on ultimate conditions of quasi-static compression tests. (**a**) Elastic modulus. (**b**) Poisson’s ratio.

**Figure 14 materials-16-04947-f014:**
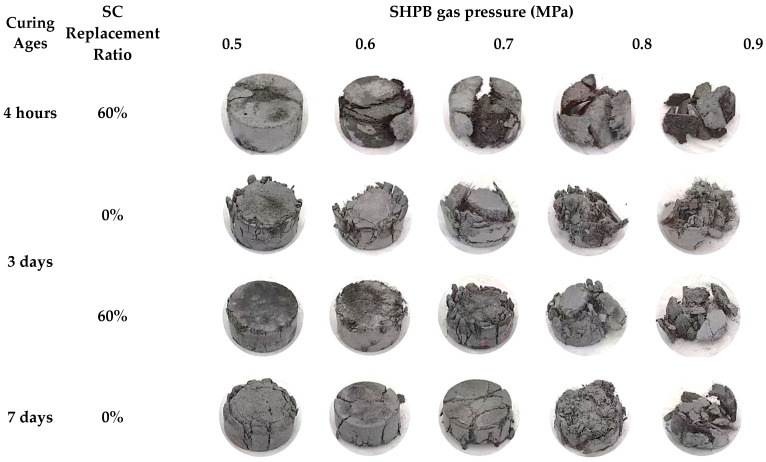
Typical failure modes of HSFRC specimens under dynamic compression tests.

**Figure 15 materials-16-04947-f015:**
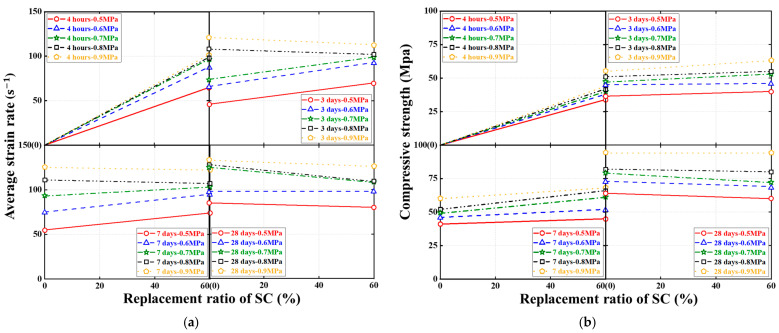
Effect of SC replacement ratios on ultimate conditions of dynamic compression tests. (**a**) Average strain rate. (**b**) Compressive strength. (**c**) Ultimate axial strain.

**Figure 16 materials-16-04947-f016:**
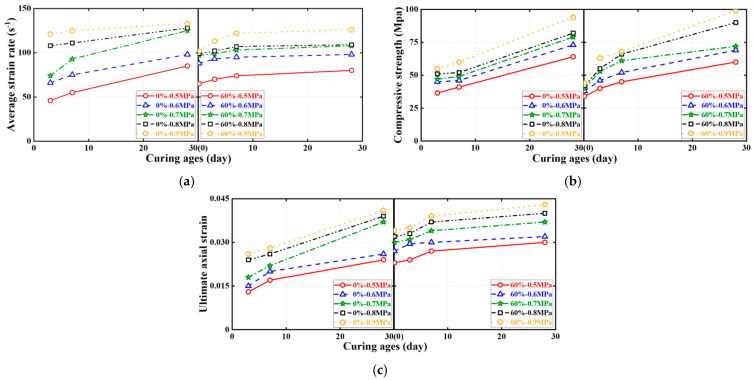
Effect of concrete curing ages on ultimate conditions of dynamic compression tests. (**a**) Average strain rate. (**b**) Compressive strength. (**c**) Ultimate axial strain.

**Figure 17 materials-16-04947-f017:**
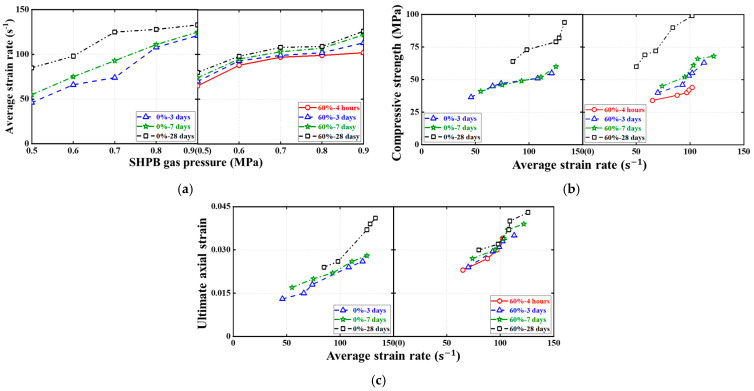
Effect of strain rates on ultimate conditions of dynamic compression tests. (**a**) Average strain rate. (**b**) Compressive strength. (**c**) Ultimate axial strain.

**Figure 18 materials-16-04947-f018:**
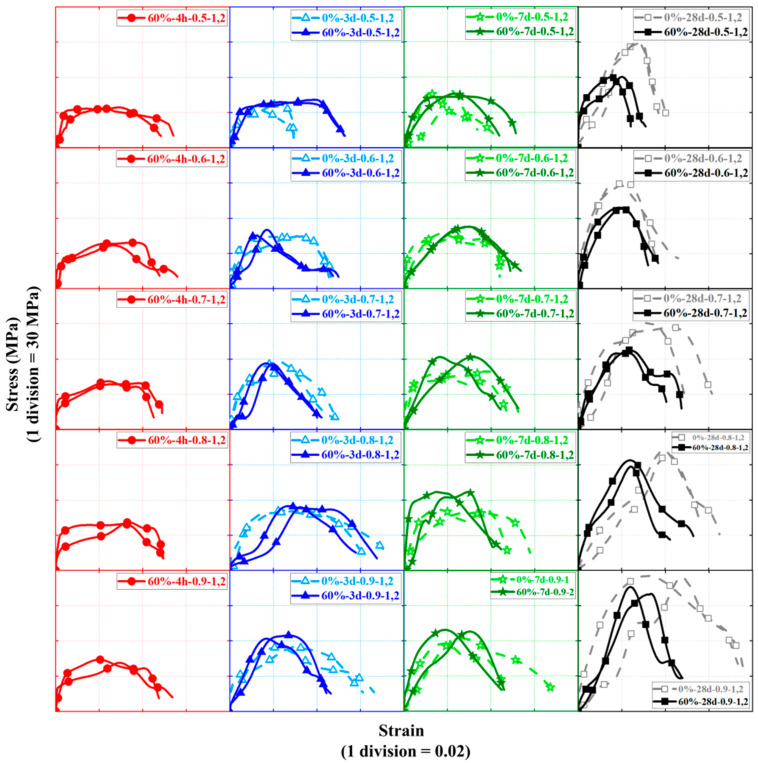
Stress–strain curves of dynamic compression tests.

**Table 1 materials-16-04947-t001:** Detailed design of test specimens.

Specimen ID	Test Type	Specimen Type	SC Replacement Ratio	Curing Age	Diameter × Height (Side Length)	SHPB Gas Pressure
CQC-0%-3d-1,-2,-3	Quasi- static	Cube	0%	3 days	100 mm	--
CQC-0%-7d-1,-2,-3				7 days	100 mm	
CQC-0%-28d-1,-2,-3				28 days	100 mm	
CQC-60%-4h-1,-2,-3			60%	4 h	100 mm	
CQC-60%-3d-1,-2,-3				3 days	100 mm	
CQC-60%-7d-1,-2,-3				7 days	100 mm	
CQC-60%-28d-1,-2,-3				28 days	100 mm	
QC-0%-3d-1,-2,-3		Cylinder	0%	3 days	100 mm × 200 mm	
QC-0%-7d-1,-2,-3				7 days	100 mm × 200 mm	
QC-0%-28d-1,-2,-3				28 days	100 mm × 200 mm	
QC-60%-4h-1,-2,-3			60%	4 h	100 mm × 200 mm	
QC-60%-3d-1,-2,-3				3 days	100 mm × 200 mm	
QC-60%-7d-1,-2,-3				7 days	100 mm × 200 mm	
QC-60%-28d-1,-2,-3				28 days	100 mm × 200 mm	
DC-0%-3d-0.5-1,-2	Dynamic	FBD	0%	3 days	100 mm × 50 mm	0.5
DC-0%-3d-0.6-1,-2					100 mm × 50 mm	0.6
DC-0%-3d-0.7-1,-2					100 mm × 50 mm	0.7
DC-0%-3d-0.8-1,-2					100 mm × 50 mm	0.8
DC-0%-3d-0.9-1,-2					100 mm × 50 mm	0.9
DC-0%-7d-0.5-1,-2				7 days	100 mm × 50 mm	0.5
DC-0%-7d-0.6-1,-2					100 mm × 50 mm	0.6
DC-0%-7d-0.7-1,-2					100 mm × 50 mm	0.7
DC-0%-7d-0.8-1,-2					100 mm × 50 mm	0.8
DC-0%-7d-0.9-1,-2					100 mm × 50 mm	0.9
DC-0%-28d-0.5-1,-2				28 days	100 mm × 50 mm	0.5
DC-0%-28d-0.6-1,-2					100 mm × 50 mm	0.6
DC-0%-28d-0.7-1,-2					100 mm × 50 mm	0.7
DC-0%-28d-0.8-1,-2					100 mm × 50 mm	0.8
DC-0%-28d-0.9-1,-2					100 mm × 50 mm	0.9
DC-60%-4h-0.5-1,-2			60%	4 h	100 mm × 50 mm	0.5
DC-60%-4h-0.6-1,-2					100 mm × 50 mm	0.6
DC-60%-4h-0.7-1,-2					100 mm × 50 mm	0.7
DC-60%-4h-0.8-1,-2					100 mm × 50 mm	0.8
DC-60%-4h-0.9-1,-2					100 mm × 50 mm	0.9
DC-60%-3d-0.5-1,-2				3 days	100 mm × 50 mm	0.5
DC-60%-3d-0.6-1,-2					100 mm × 50 mm	0.6
DC-60%-3d-0.7-1,-2					100 mm × 50 mm	0.7
DC-60%-3d-0.8-1,-2					100 mm × 50 mm	0.8
DC-60%-3d-0.9-1,-2					100 mm × 50 mm	0.9
DC-60%-7d-0.5-1,-2				7 days	100 mm × 50 mm	0.5
DC-60%-7d-0.6-1,-2					100 mm × 50 mm	0.6
DC-60%-7d-0.7-1,-2					100 mm × 50 mm	0.7
DC-60%-7d-0.8-1,-2					100 mm × 50 mm	0.8
DC-60%-7d-0.9-1,-2					100 mm × 50 mm	0.9
DC-60%-28d-0.5-1,-2				28 days	100 mm × 50 mm	0.5
DC-60%-28d-0.6-1,-2					100 mm × 50 mm	0.6
DC-60%-28d-0.7-1,-2					100 mm × 50 mm	0.7
DC-60%-28d-0.8-1,-2					100 mm × 50 mm	0.8
DC-60%-28d-0.9-1,-2					100 mm × 50 mm	0.9

**Table 2 materials-16-04947-t002:** Key results of quasi-static compression tests.

Specimen ID	Compressive Strength (MPa)	Ultimate Axial Strain	Elastic Modulus (GPa)	Poisson’s Ratio
CQC-0%-3d-1,-2,-3	72.62	--	--	--
CQC-0%-7d-1,-2,-3	90.65	--	--	--
CQC-0%-28d-1,-2,-3	124.89	--	--	--
CQC-60%-4h-1,-2,-3	45.16	--	--	--
CQC-60%-3d-1,-2,-3	52.56	--	--	--
CQC-60%-7d-1,-2,-3	62.55	--	--	--
CQC-60%-28d-1,-2,-3	80.94	--	--	--
QC-0%-3d-1,-2,-3	49.52	0.0022	33.1	0.15
QC-0%-7d-1,-2,-3	56.17	0.0020	37.0	0.19
QC-0%-28d-1,-2,-3	90.95	0.0018	43.6	0.21
QC-60%-4h-1,-2,-3	33.14	0.0046	26.5	0.19
QC-60%-3d-1,-2,-3	39.14	0.0030	27.6	0.20
QC-60%-7d-1,-2,-3	54.06	0.0028	34.1	0.21
QC-60%-28d-1,-2,-3	70.62	0.0023	38.3	0.22

**Table 3 materials-16-04947-t003:** Key results of dynamic compression tests.

Specimen ID	Average Strain Rate $\left(s^{-1}\right)$	Compressive Strength (MPa)	Ultimate Axial Strain
DC-0%-3d-0.5-1	43	35.56	0.013
DC-0%-3d-0.5-2	49	37.82	0.021
DC-0%-3d-0.6-1	66	45.32	0.012
DC-0%-3d-0.6-2	67	47.99	0.015
DC-0%-3d-0.7-1	71	45.84	0.019
DC-0%-3d-0.7-2	76	46.34	0.017
DC-0%-3d-0.8-1	119	50.98	0.024
DC-0%-3d-0.8-2	122	50.63	0.030
DC-0%-3d-0.9-1	127	54.20	0.022
DC-0%-3d-0.9-2	132	54.79	0.030
DC-0%-7d-0.5-1	51	40.35	0.012
DC-0%-7d-0.5-2	59	43.21	0.022
DC-0%-7d-0.6-1	72	46.72	0.023
DC-0%-7d-0.6-2	79	46.79	0.017
DC-0%-7d-0.7-1	89	47.76	0.016
DC-0%-7d-0.7-2	97	49.63	0.037
DC-0%-7d-0.8-1	102	49.08	0.036
DC-0%-7d-0.8-2	104	51.89	0.016
DC-0%-7d-0.9-1	110	57.85	0.019
DC-0%-7d-0.9-2	123	62.19	0.027
DC-0%-28d-0.5-1	80	63.68	0.027
DC-0%-28d-0.5-2	90	65.74	0.029
DC-0%-28d-0.6-1	97	72.84	0.024
DC-0%-28d-0.6-2	100	74.13	0.018
DC-0%-28d-0.7-1	125	78.06	0.043
DC-0%-28d-0.7-2	126	80.89	0.032
DC-0%-28d-0.8-1	124	81.34	0.040
DC-0%-28d-0.8-2	135	82.06	0.039
DC-0%-28d-0.9-1	128	93.81	0.047
DC-0%-28d-0.9-2	133	95.34	0.032
DC-60%-4h-0.5-1	65	33.42	0.022
DC-60%-4h-0.5-2	70	34.41	0.029
DC-60%-4h-0.6-1	83	37.07	0.026
DC-60%-4h-0.6-2	96	39.18	0.036
DC-60%-4h-0.7-1	97	39.43	0.038
DC-60%-4h-0.7-2	98	41.08	0.024
DC-60%-4h-0.8-1	98	39.52	0.031
DC-60%-4h-0.8-2	99	40.99	0.033
DC-60%-4h-0.9-1	100	42.32	0.029
DC-60%-4h-0.9-2	104	43.97	0.020
DC-60%-3d-0.5-1	70	39.04	0.023
DC-60%-3d-0.5-2	71	40.82	0.039
DC-60%-3d-0.6-1	93	44.85	0.032
DC-60%-3d-0.6-2	95	46.73	0.032
DC-60%-3d-0.7-1	98	51.41	0.026
DC-60%-3d-0.7-2	100	55.89	0.022
DC-60%-3d-0.8-1	100	53.92	0.032
DC-60%-3d-0.8-2	103	54.86	0.027
DC-60%-3d-0.9-1	105	61.82	0.017
DC-60%-3d-0.9-2	122	64.53	0.028
DC-60%-7d-0.5-1	63	43.33	0.029
DC-60%-7d-0.5-2	66	45.83	0.024
DC-60%-7d-0.6-1	80	52.75	0.029
DC-60%-7d-0.6-2	83	52.91	0.030
DC-60%-7d-0.7-1	98	61.82	0.017
DC-60%-7d-0.7-2	103	61.93	0.032
DC-60%-7d-0.8-1	107	67.01	0.030
DC-60%-7d-0.8-2	107	67.08	0.015
DC-60%-7d-0.9-1	134	67.82	0.030
DC-60%-7d-0.9-2	111	69.21	0.021
DC-60%-28d-0.5-1	80	60.13	0.029
DC-60%-28d-0.5-2	82	60.40	0.030
DC-60%-28d-0.6-1	98	68.43	0.032
DC-60%-28d-0.6-2	99	69.36	0.033
DC-60%-28d-0.7-1	106	70.03	0.036
DC-60%-28d-0.7-2	110	71.63	0.038
DC-60%-28d-0.8-1	108	88.54	0.038
DC-60%-28d-0.8-2	110	93.89	0.042
DC-60%-28d-0.9-1	123	99.06	0.043
DC-60%-28d-0.9-2	126	99.86	0.044

## Data Availability

Not applicable.

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
