# Peer review of "Rapid-Hardening and High-Strength Steel-Fiber-Reinforced Concrete: Effects of Curing Ages and Strain Rates on Compressive Performance"

_materials, 2023, doi:10.3390/ma16144947_

Round 1

Reviewer 1 Report

This paper presents an experimental study of the quasi-static and dynamic compression behavior of conventional and fast-curing HSFRCs. variable parameters in the tests were the time of concrete maturation, e.g. after 4 hours and classically after 7 and 28 days of maturation, and the impact of strain rate.

The experiments carried out are well described and explained and the conclusions are correct.

In my opinion, in order to fully assess the properties of HSFRC high-speed concrete, it is necessary to supplement the tests with:

- determination of the skin,

- determination of durability under the action of water and frost,

- alkaline reactivity of HSFRC concrete with new cement.

after completing the supplement, I will ask the authors to re-send the paper in order to verify the entirety of the research.

Author Response

Detail responses to the reviewer‘s comments have been provided in an attachment named “Response to Reviewer 1”.

Reviewer 2 Report

Figure 1 shows not only the expansion joint, but also the work carried out during the repair

In section 2.1, it is not clear whether it is a mass or volume ratio

Section 2.3: Under what conditions were the specimens cured?

Figure 6: What typical cracks were observed? how are they different

why does Figure 8 compare CQC-0 to QC-60 and not CQC-0 to QC-0?

or CQC-60 with QC-60? do their meanings match?

Maybe formula 2 and 3 should be under research methods?

The English language is understandable and not complicated, there are a lot of abbreviations, and it is not clear what CQC, QC, DC means, because there is no clear explanation

Author Response

Detail responses to the reviewer‘s comments have been provided in an attachment named “Response to Reviewer 2”.

Reviewer 3 Report

Thank you for this contribution. This is an interesting and timely manuscript. This paper discusses how novel machine learning can be used in earthquake problems on corroded RC columns. The conducted analysis is typically standard and falls within the expected work from such a publication and hence the work merits publication. As such, the authors are invited to properly address the following items:

1. In general, the introduction is light and does not represent state of the art in this domain. The amount of work in this area continues to rise rapidly. The authors are advised to strengthen their literature review section with supplementary material. Perhaps the addition of 1-2 pages can help strengthen this section.

2. What standard testing did the authors follow? Please report and cite such standards. 

3. In Fig. 13, what do the failure modes tell us about the specimens?

4. The conclusion section should include main highlight, main findings, limitations and areas of future research on this study. Ensure that all nomenclature and all symbols and abbreviations are well defined.

Author Response

Detail responses to the reviewer‘s comments have been provided in an attachment named “Response to Reviewer 3”.

Round 2

Reviewer 1 Report

I would like to thank the authors for their specific responses to the questions arising from the review of the work and after reading the answers, I believe that the article can be published in the journal.

Reviewer 3 Report

.